# Learning Interpolations between Boltzmann Densities

**Bálint Máté**                                                  *balint.mate@unige.ch*
*Department of Computer Science, Department of Physics*
*University of Geneva*

**François Fleuret**                                             *francois.fleuret@unige.ch*
*Department of Computer Science*
*University of Geneva*

**Reviewed on OpenReview:** *https://openreview.net/forum?id=TH6YrEcbth*

## Abstract

We introduce a training objective for continuous normalizing flows that can be used in the absence of samples but in the presence of an energy function. Our method relies on either a prescribed or a learnt interpolation $f_t$ of energy functions between the target energy $f_1$ and the energy function of a generalized Gaussian $f_0(x) = ||x/\sigma||_p^p$. The interpolation of energy functions induces an interpolation of Boltzmann densities $p_t \propto e^{-f_t}$ and we aim to find a time-dependent vector field $V_t$ that transports samples along the family $p_t$ of densities. The condition of transporting samples along the family $p_t$ is equivalent to satisfying the continuity equation with $V_t$ and $p_t = Z_t^{-1} e^{-f_t}$. Consequently, we optimize $V_t$ and $f_t$ to satisfy this partial differential equation. We experimentally compare the proposed training objective to the reverse KL-divergence on Gaussian mixtures and on the Boltzmann density of a quantum mechanical particle in a double-well potential.

## 1 Introduction

We consider the task of estimating the expectation value $\mathbb{E}_{x \sim p}[\mathcal{O}(x)]$ of some observable $\mathcal{O}$, under a probability density $p$ proportional to the unnormalized density $e^{-f}$, where $f : \mathbb{R}^n \to \mathbb{R}$ is a given energy function. In particular, we don't have access to true samples from $p$, all we have is the ability to evaluate $f$ and its derivatives for any $x \in \mathbb{R}^n$. A popular technique (Boyda et al., 2021; Albergo et al., 2021a;b; 2022; Abbott et al., 2022; de Haan et al., 2021; Gerdes et al., 2022; Noé et al., 2018; Köhler et al., 2020; Nicoli et al., 2020; 2021) for attacking this problem is to use a normalizing flow to parametrize a variational density $q_\theta$ and optimize the parameters $\theta$ to minimize the reverse KL-divergence

$$KL[q_\theta, p] = \mathbb{E}_{x \sim q_\theta}(\log q_\theta(x) - \log p(x)) = \mathbb{E}_{x \sim q_\theta}(\log q_\theta(x) + f(x)) + Z. \tag{1}$$

The use of normalizing flows for this problem is particularly attractive because $q_\theta$ can be used as a proposal for importance sampling, $\mathbb{E}_{x \sim p}[\mathcal{O}(x)] = E_{x \sim q_\theta}[\frac{p(x)}{q_\theta(x)}\mathcal{O}(x)]$, to account for the inaccuracies of $q_\theta$. Unfortunately, the reverse KL-divergence is mode-seeking, making the training prone to mode-collapse (Fig. 2). To tackle this problem, several works have proposed alternative training objectives for normalizng flows. Vaitl et al. (2022) introduce better estimators of the forward and reverse KL divergences, while Midgley et al. (2022) use the $\alpha = 2$ divergence instead of the reverse KL-divergence as their training objective. We propose yet another alternative based on infinitesimal deformations of Boltzmann densities (Pfau & Rezende, 2020; Máté & Fleuret, 2022). This work was motivated by denoising diffusion (Sohl-Dickstein et al., 2015; Ho et al., 2020) and score-base models (Song & Ermon, 2019). It also bears similarities to more recent works (Lipman et al., 2022; Albergo & Vanden-Eijnden, 2022; Liu et al., 2022; Neklyudov et al., 2022) that generalize diffusion models by relying on more flexible interpolations between the data and the base distribution.

---

The implementation of our experiments is available at `https://github.com/balintmate/boltzmann-interpolations`.

The contributions of this work can be summarized as follows

- In §3 we describe our method which relies on either a prescribed or a learnt interpolation $f_t$ of energy functions between the target energy $f_1$ and the energy function of a generalized Gaussian $f_0(x) = ||x/\sigma||_p^p$. Given $f_t$ we optimize a vector field $V_t$ to transport samples along the family $p_t(x) \propto e^{-f_t(x)}$ of Boltzmann densities. After translating this condition to a PDE between $V_t$ and $f_t$ we propose to minimize the amount by which this PDE fails to hold.
  - First we find that in certain cases the linear interpolation $f_t = (1-t)f_0 + tf_1$ already leads to improved performance over optimizing the reverse KL-divergence. We also show that in general this interpolation is insufficient.
  - Motivated by the failure mode of the linear interpolation, we parametrize the interpolation with another neural network $f_t = (1-t)f_0 + tf_1 + t(1-t)f^\theta(t)$ and optimize $f_t^\theta$ together with the vector field $V_t^\theta$.
- In §4 we run experiments on Gaussian mixtures and on the Boltzmann density of a quantum particle in a double-well potential, and report improvements in KL-divergence, effective sample size, mode coverage and also training speed.

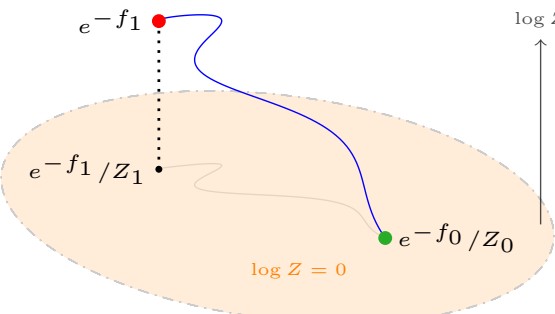

Figure 1: Interpolating in the space of unnormalized probability densities. The unnormalized target density (red) and its normalization (black) that we are trying to fit a flow to. The base density of the flow (green) lying on the subspace of normalized probability densities (orange). The interpolation connecting the base density to the unnormalized target (blue) and the induced interpolation of normalized densities (the projection of the blue trajectory to the "$\log Z = 0$"-plane).

## Motivation

Consider the following multimodal density

$$\frac{1}{4}\Big(\mathcal{N}\left(\begin{bmatrix} -8 & -8 \end{bmatrix}, 1\right) + \mathcal{N}\left(\begin{bmatrix} -8 & 8 \end{bmatrix}, 1\right) + \mathcal{N}\left(\begin{bmatrix} 8 & -8 \end{bmatrix}, 1\right) + \mathcal{N}\left(\begin{bmatrix} 8 & 8 \end{bmatrix}, 1\right)\Big), \tag{2}$$

where $\mathcal{N}(\mu, \sigma)$ denotes a normal density centered at $\mu$ with covariance matrix $\mathrm{diag}(\sigma^2)$. Fig. 2 shows the mode-collapse of a normalizing flow trained with the reverse KL-divergence on this target. The reason why mode collapse can happen in the first place is the training objective itself. The mode seeking behavior of the reverse KL-divergence can be explained as follows. As the sampling is done according to $q_\theta$, the difference in log-likelihoods is weighted by the likelihood $q_\theta$. This implies that if $q_\theta$ completely ignores modes of $p$ the log-likelihoods of $p$ and $q_\theta$ are not compared over regions that are not covered by $q_\theta$. In this paper we introduce a training objective for continuous normalizing flows that can be used to replace the reverse KL-divergence and investigate to what extent it solves the issue of mode collapse on multimodal targets.

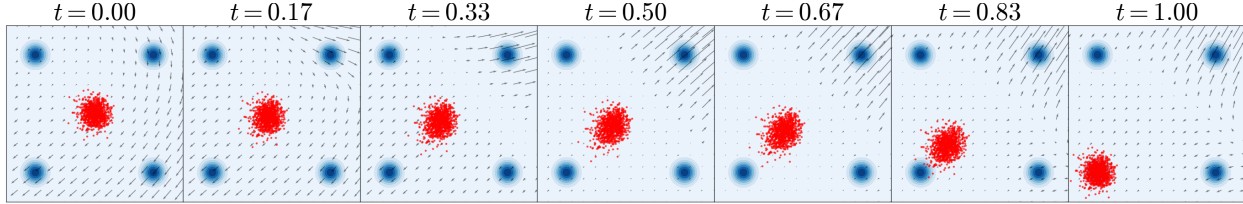

Figure 2: Mode-seeking nature of the reverse KL-divergence. The figures from left to right show how the latent gaussian is transformed by the continuous normalizing flow trained with the reverse KL objective. The blue blobs denote the target density (2), the arrows represent the vector field $V_t$.

## 2   Background

**Change of variables.**   Let $p_0(z)$ be a probability density on $\mathbb{R}^n$ and $\Psi : \mathbb{R}^n \to \mathbb{R}^n$ a diffeomorphism. Pushing the density $p_0$ forward along $\Psi$ induces a new probability density $p$ implicitly defined by

$$\log p_0(z) = \log p(\Psi z) + \log |\det J_\Psi(z)|. \tag{3}$$

The term $\log |\det J_\Psi(z)|$ measures how much the function $\Psi$ expands volume locally at $z$.

**Continuous change of variables.**   Let now $V_t$ be a time-dependent vector field and $\Psi_\tau$ denote the diffeomorphism of flowing along the integral curves of $V_t$ from 0 to $\tau$. This family of diffeomorphisms generates a one-parameter family of densities $p_\tau$. The amount of volume expansion a particle experiences along a trajectory $t \mapsto \Phi_t z$ between 0 and $\tau$ is $\int_0^\tau \nabla \cdot V_t(\Psi_t z) dt$. The log-likelihoods are then related by

$$\log p_0(z) = \log p_\tau(\Psi_\tau z) + \int_0^\tau \nabla \cdot V_t(\Psi_t z) dt. \tag{4}$$

**Normalizing flows.**   Normalizing flows (Tabak & Turner, 2013; Rezende & Mohamed, 2015; Dinh et al., 2016) (continuous normalizing flows (Chen et al., 2018)) parametrize a subset of the space of all densities on $\mathbb{R}^n$. They do this by first fixing a base density $p_0$ and using a neural network that parametrizes the transformation $\Psi$ (the vector field $V_t$). The (continuous) change of variables formula is then applied to compute the density induced by $\Psi$ ($V_t$). Recently, generalizations of normalizing flows have also been studied (Nielsen et al., 2020; Huang et al., 2020; Máté et al., 2022).

**Boltzmann densities.**   Let $f : \mathbb{R}^n \to \mathbb{R}$ be an energy function with a finite normalizing constant $Z = \int e^{-f(x)} d^n x$. The function $f$ then induces a Boltzmann density over the configurations $x \in \mathbb{R}^n$, $p(x) = \frac{1}{Z} e^{-f(x)}$. Conversely, given a probability density function $p : \mathbb{R}^n \to \mathbb{R}_{+,0}$ the corresponding energy function can be recovered up to a constant $f = -\log p - \log Z$.

**The continuity equation.**   One can describe a time-dependent probability density either with its time dependent density function $p_t$ or by a reference (base) density $p_0$ and a time-dependent vector field $V_t$. The latter generates $p_\tau$ by pushing $p_0$ forward along the integral curves of $V_t$ from $t = 0$ to $t = \tau$. The two descriptions are related by the continuity equation $\partial_t p_t + \nabla \cdot (p_t V_t) = 0$, which, in the case of Boltzmann densities $p_t = Z_t^{-1} e^{-f_t}$ can be rewritten as follows,

$$
\begin{aligned}
0 &= \partial_t p_t + \nabla \cdot (p_t V_t) && \text{expanding } p_t = Z_t^{-1} e^{-f_t} && (5)\\
&= \partial_t(Z_t^{-1} e^{-f_t}) + \nabla \cdot (Z_t^{-1} e^{-f_t} V_t) && && (6)\\
&= e^{-f_t} \partial_t(Z_t^{-1}) + Z_t^{-1} \partial_t(e^{-f_t}) + Z_t^{-1} \nabla \cdot (e^{-f_t} V_t) && && (7)\\
&= e^{-f_t} \partial_t(Z_t^{-1}) + Z_t^{-1} \partial_t(e^{-f_t}) + Z_t^{-1} e^{-f_t} \nabla \cdot V_t + Z_t^{-1} \langle \nabla e^{-f_t}, V_t \rangle && && (8)\\
&= e^{-f_t} \partial_t(Z_t^{-1}) - Z_t^{-1} e^{-f_t} \partial_t(f_t) + Z_t^{-1} e^{-f_t} \nabla \cdot V_t - Z_t^{-1} e^{-f_t} \langle \nabla f_t, V_t \rangle && \text{factoring out } p_t = Z_t^{-1} e^{-f_t} && (9)\\
&= Z_t^{-1} e^{-f_t} \Big( \underbrace{Z_t \partial_t(Z_t^{-1})}_{-\partial_t \log Z_t} - \partial_t f_t + \nabla \cdot V_t - \langle \nabla f_t, V_t \rangle \Big), && && (10)
\end{aligned}
$$

where $\langle \, , \, \rangle$ is the Euclidean scalar product between vectors. Since $p_t = Z_t^{-1} e^{-f_t} > 0$, we conclude

$$\partial_t f_t + \langle \nabla f_t, V_t \rangle - \nabla \cdot V_t + \partial_t \log Z_t = 0, \tag{11}$$

**Lemma 1.** *Moreover, if any time-dependent energy function $f_t$, time-dependent vector field $V_t$ and spatially constant function $C_t$ satisfies*

$$\partial_t f_t + \langle \nabla f_t, V_t \rangle - \nabla \cdot V_t + C_t = 0, \tag{12}$$

*then $C_t$ necessarily equals $\partial_t \log Z_t$ with $Z_t = \int e^{-f_t(x)} d^n x$.*

*Proof.* Let $f_t, V_t, C_t$ be such that they satisfy (12). We can then compute

$$\partial_t \log Z_t = \frac{\partial_t Z_t}{Z_t} = \frac{\int (-\partial_t f_t) e^{-f_t(x)} d^n x}{\int e^{-f_t(x)} d^n x} = \frac{\int \overbrace{(\langle \nabla f_t, V_t \rangle - \nabla \cdot V_t) e^{-f_t(x)}}^{-\nabla \cdot (e^{-f_t(x)} V_t)} d^n x}{\int e^{-f_t(x)} d^n x} + \frac{\int C_t e^{-f_t(x)} d^n x}{\int e^{-f_t(x)} d^n x} = C_t \quad (13)$$

where the last equality follows from the divergence theorem and the fact that $e^{-f_t(x)}$ vanishes at infinity. $\square$

When numerically solving the continuity equation, Lemma 1 enables us to work with (12) instead of (11) and parametrize the triplet $(f, V, C)$ with $C$ being a spatially constant function instead of estimating $\partial_t \log Z_t$. In what follows we only work with (12) and refer to it as the *continuity equation*.

## 3 Approximating the transport field

From here on, we will use the vector field $V_t$ and the term "continuous normalizing flow" interchangeably. Our goal is to sample from a target Boltzmann density $p_{\text{target}} \propto e^{-f_{\text{target}}}$ by

1. defining a family of energy functions $f_t$, $0 \le t \le 1$ interpolating between the target energy $f_1 = f_{\text{target}}$ and the energy function of a generalized Gaussian $f_0(x) = ||x/\sigma||_p^p$,

2. finding a transport field $V_t$ such that $(f_t, V_t)$ "solves" the continuity equation (12).

If we succeed at both of these constructions, then we can obtain samples from $p_{\text{target}}$ by sampling from $p_0 \propto e^{-||x/\sigma||_p^p}$ and let the samples follow the integral curves of $V_t$ from $t = 0$ to $t = 1$. Note that we turned the problem of learning a single density $e^{-f_1}$ into a continuous collection of problems, learning all the densities $e^{-f_t}$ for $0 \le t \le 1$. It might seem like that we just made the task more difficult, but the idea is that for any $0 \le \tau \le 1$, the density $e^{-f_\tau}$ is easier to fit once $e^{-f_{\tau-\varepsilon}}$ is already fitted for some small $\varepsilon$.

**The pointwise continuity error**

Regarding the second item of the above list, an analytical expression for $V_t$ is not easy to find if we are given a family of energy functions $f_t$. This would amount to solving (12), which is difficult in general. Therefore we will parametrize $V_t$ with a neural network and train it to minimize the amount by which the pair $(f_t, V_t)$ fails to satisfy the continuity equation. We begin by recalling the continuity equation,

$$\partial_t f_t + \langle \nabla f_t, V_t \rangle - \nabla \cdot V_t + C_t = 0, \quad (14)$$

where $C_t$ is a spatially constant function. In what follows, $V_t^\theta$ and $C_t^\theta$ are parametrized by neural networks and are trained to minimize some monotonically increasing function $L$[1] of the pointwise *continuity error*

$$\mathcal{E}_{\theta,x,t} = |\partial_t f_t(x) + \langle \nabla f_t(x), V_t^\theta(x) \rangle - \nabla \cdot V_t^\theta(x) + C_t^\theta|. \quad (15)$$

The expression $L(\mathcal{E}_{\theta,x,t})$ measures the incompatibility of $f_t$ and $V_t$ at a single $(t, x)$ pair of coordinates, we will need to optimize some sort of integral of this pointwise error over both $t$ and $x$.

**The continuity loss**

Suppose that we have an interpolation of energy functions $f_t$. We propose to train $V_t^\theta$ and $C_t^\theta$ to minimize the continuity error (15) along the trajectories of $V_t^\theta$. Formally, let $q_\theta$ be a parametric density parametrized by a continuous normalizing flow $V_t^\theta$. We update the parameters to minimalize the integral of $L(\mathcal{E})$ along the trajectories of the flow,

$$\mathcal{L}(\theta) = \mathbb{E}_{z \sim p_0} \left[ \int_0^1 L\left( \mathcal{E}_{\theta, \gamma_t^\theta(z), t} \right) dt \right], \quad (16)$$

where $\gamma_\tau^\theta(z)$ is given by transporting $z$ along the vector field $V_t^\theta$ between 0 and $\tau$. We evaluate the integral (16) by discretizing time and using numerical ODE solvers.

---

[1] In our experiments we tried $L : \mathbb{R}^+ \to \mathbb{R}^+ \in \{\mathcal{E} \mapsto \mathcal{E}, \mathcal{E} \mapsto \mathcal{E}^2, \mathcal{E} \mapsto \mathcal{E} + \mathcal{E}^2, \mathcal{E} \mapsto \log(1 + \mathcal{E})\}$.

**First approach: Linear interpolation**

Arguably the simplest way to interpolate between a pair of functions $(f_0, f_1)$ is to set

$$f_t(x) = (1-t)f_0(x) + tf_1(x), \qquad 0 \le t \le 1. \tag{17}$$

The same interpolation is proposed by Wu et al. (2020) in the context of stochastic normalizing flows and is also a common choice for annealed importance sampling (Neal, 1998). We will call the family (17) the *linear interpolation*. Fig. 3 shows the evolution of the samples along a continuous normalizing flow trained with the continuity loss using the linear interpolation. We observe that the same network trained with the continuity loss instead of the reverse KL-divergence can capture all 4 modes of (2).

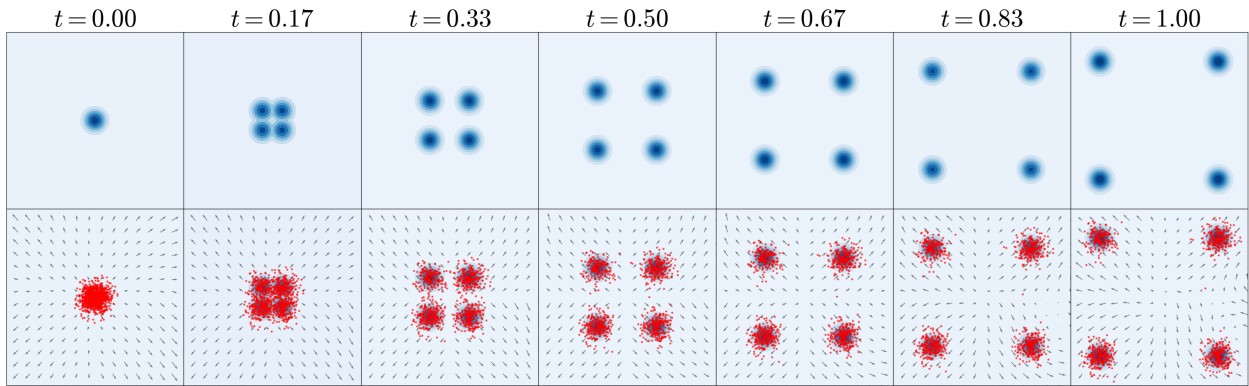

Figure 3: The same continuous normalizing flow as in Fig. 2 trained with the continuity loss using the linear interpolation. The top row shows how the target density evolves under the predefined linear interpolation between $f_0 = x^2/2$ and $f_1 = -\log(\text{Eq. 2})$, while the bottom row shows how the samples from $q_\theta$ evolve along $V_t$ as $t$ is varied.

**The issue with the linear interpolation.** Let us now consider the density

$$\tfrac{1}{3}\mathcal{N}\left(\begin{bmatrix} 4 & 4 \end{bmatrix}, 1\right) + \tfrac{2}{3}\mathcal{N}\left(\begin{bmatrix} -8 & -8 \end{bmatrix}, 1\right). \tag{18}$$

This target does not enjoy the symmetry properties of the previous mixture, one of the modes is closer to the base and has a lower relative weight than the other one. The top row of Fig. 4 shows the linear interpolation to this target and the second row shows that this interpolation is insufficient to capture the mode which is further away. In a nutshell, the reason for this is that the linear interpolation does not preserve the relative weights of the modes as $t$ is varied.

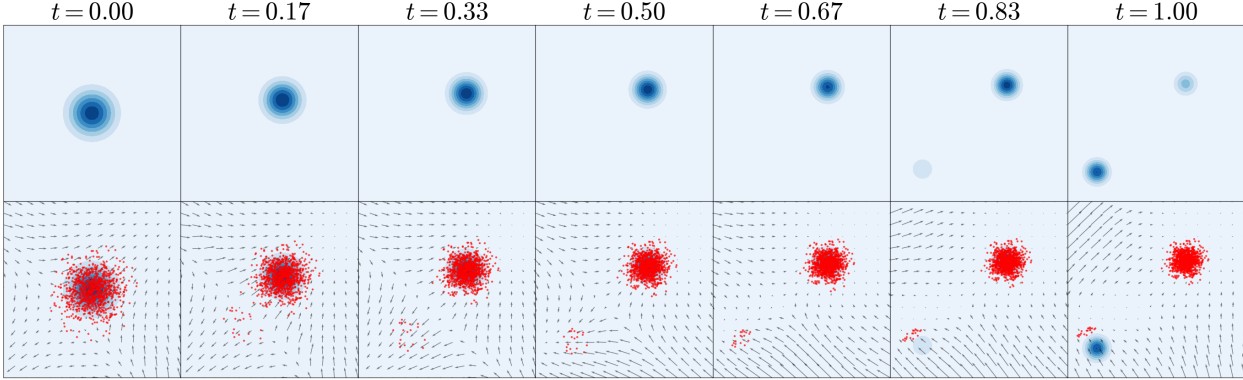

Figure 4: The issue with the linear interpolation. The top row shows how the target density evolves under the predefined linear interpolation between $f_0 = x^2/8$ and $f_1 = -\log(\text{Eq. 18})$, while the bottom row shows how the samples from $q_\theta$ evolve along $V_t$ as $t$ is varied.

**Interlude: good and bad interpolations**

The reason why the linear interpolation can lead to problems is illustrated in Figures 5 and 6. Figure 5 shows a particular example where the linear interpolation in log-density space induces a "good" interpolation in density space. Figure 6 shows that for certain target functions the linear interpolation in log-density space induces an interpolation in density space that is not "local" in the sense that probability mass gets moved between the modes. Such densities are difficult to learn with the linear interpolation, even if in principle there exists a vector field $V_t$ that moves probability mass the correct way along any interpolation.

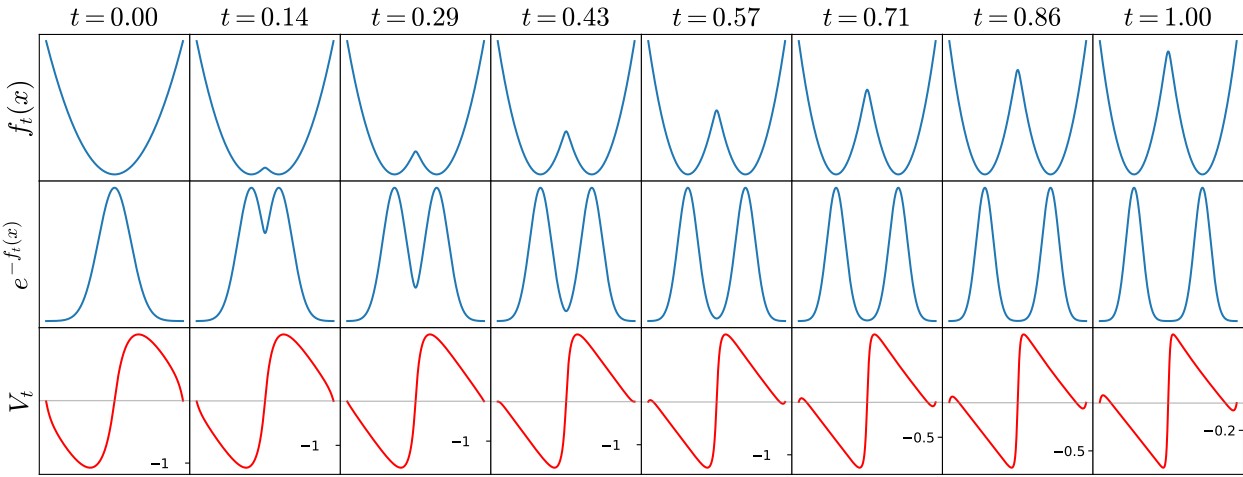

Figure 5: Linear interpolation between $f_0(x) = x^2/4$ and $f_1 = -\log(\frac{1}{2}e^{-(x-3)^2} + \frac{1}{2}e^{-(x+3)^2})$. Linear interpolation in the log-density space (top row), and the induced interpolation in density space (middle row). The transport field $V_t$ which can be represented as a scalar-valued function for one-dimensional densities (bottom row). The thin gray line in the bottom row denotes $V_t = 0$.

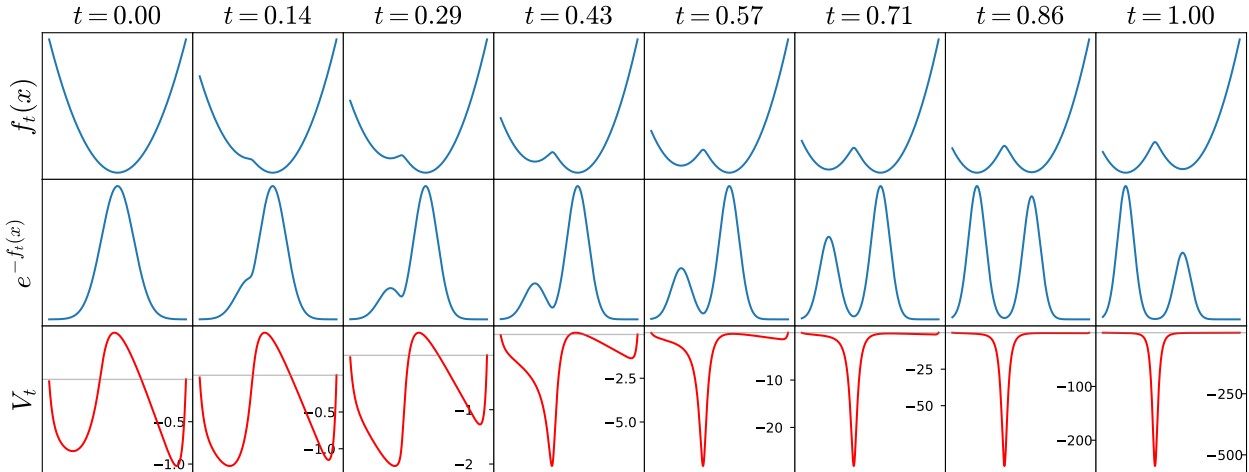

Figure 6: Linear interpolation between $f_0(x) = x^2/4$ and $f_1 = -\log(\frac{1}{3}e^{-(x-1)^2} + \frac{2}{3}e^{-(x+4)^2})$. Linear interpolation in the log-density space (top row), and the induced interpolation in density space (middle row). The transport field $V_t$ which can be represented as a scalar-valued function for one-dimensional densities (bottom row). The thin gray line in the bottom row denotes $V_t = 0$. Note how the relative weights of modes is not preserved as $t$ is varied, resulting in the exploding norm of $V_t$.

**Sanity check: interpolate along the diffusion process**

**Diffusion Processes.** Diffusion models (Sohl-Dickstein et al., 2015; Ho et al., 2020) learn to reverse a diffusion process that is in essence also a family of probability densities $p_t$ on $\mathbb{R}^n$ interpolating between the data density $p_1(x)$ and the latent $n-$dimensional gaussian $p_0 = \mathcal{N}(0, \sigma)$. Samples from $p_t$ are of the form $\sqrt{t}x + \sqrt{1-t}z$ where $x \sim p_1, z \sim p_0$. Geometrically, the probability density $p_t$ is obtained by first stretching $p_1$ by a factor of $\sqrt{t}$ and then convolving with the Gaussian kernel $\mathcal{G}^{\sigma\sqrt{1-t}}$[2].

$$p_1(x) \xrightarrow{\lambda_t} t^{-n/2}p_1(t^{-1/2}x) \xrightarrow{G_t} \underbrace{t^{-n/2}p_1(t^{-1/2}x) * \mathcal{G}^{\sigma\sqrt{1-t}}}_{p_t(x)}, \tag{19}$$

where we used $\lambda_t$ to denote the stretching by a factor of $\sqrt{t}$ and $G_t$ to denote the convolution with the Gaussian kernel $\mathcal{G}^{\sigma\sqrt{1-t}}$. If $t = 1$, both $\lambda_t$ (stretching by a factor of 1) and $G_t$ (convolving with a Dirac-delta) are just the identity. If $t = 0$, then $\lambda_t$ collapses $p_1$ to a Dirac-delta at the orgin and $G_t$ convolves it with $\mathcal{N}(0, \sigma)$ implying that $p_0 = \mathcal{N}(0, \sigma)$. The attractive thing about the diffusion process is that it provides an interpolation between densities where the situation of Fig. 6 is avoided. Loosely speaking, this interpolation is "local" in a sense that that the linear interpolation was not.

The family of Gaussian mixtures is closed under the diffusion process (19), we can even explicitly compute the time-evolution of a Gaussian mixture and use it to replace the linear interpolation. Fig. 7 shows samples from a normalizing flow that was trained with the continuity loss using the hand-computed interpolation of its diffusion process.

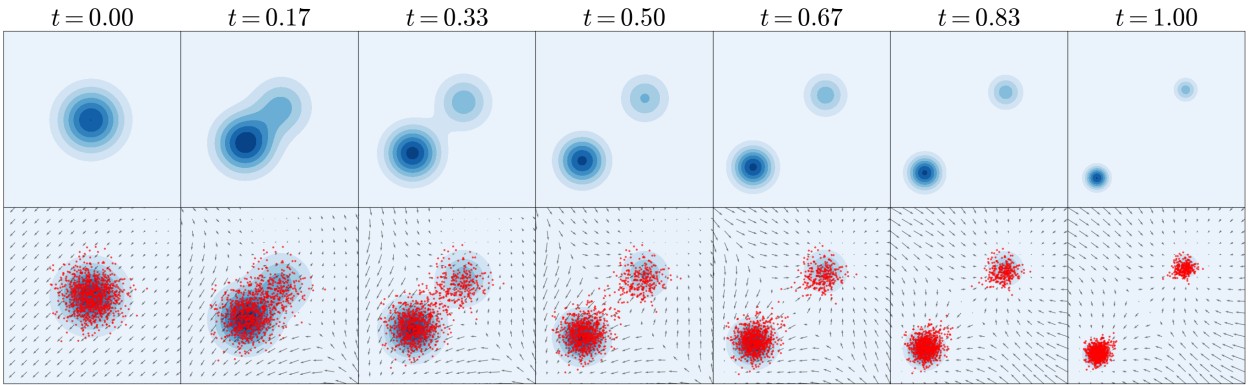

Figure 7: The same continuous normalizing flow as in Fig. 4 with the continuity loss using the diffusion interpolation. The blue blobs denote the target density (18), a mixture of two Gaussians. The top row shows how the target density evolves under the predefined diffusion-like interpolation, while the bottom row shows how the samples from $q_\theta$ evolve along the flow as $t$ is varied.

Unfortunately, since all we are given is the ability to evaluate the target energy pointwise, there is no analytical formula for calculating the diffusion process of an arbitrary target density. Moreover, any numerical approach involves integration, that gets increasingly expensive as $t$ gets smaller and the dimensionality of the problem gets larger. Nonetheless, this result demonstrates that minimizing the continuity error is a feasible approach, one just needs to be more careful about the interpolation between the latent and target energy functions. The authors could not find a predefined interpolation that is 1) "local" in the still not formal, but intuitive sense and 2) easy to compute for arbitrary target energies. Instead, we propose to learn the family of densities between the base and the target.

---

[2]The Gaussian kernel $\mathcal{G}^{\sigma\sqrt{1-t}}$ is a Gaussian density with mean 0 and covariance matrix $\text{diag}(\sigma^2(1-t))$.

**Parametrizing the interpolation**

We propose to use a neural network to parametrize the interpolation $f_t$ as

$$f_t(x) = (1-t)f_0(x) + tf_1(x) + t(1-t)f_t^\theta(x), \qquad 0 \le t \le 1, \tag{20}$$

where $f_t^\theta$ is parametrized by a neural network. This parametrization ensures that the boundary conditions at $t \in \{0,1\}$ are satisfied, and allows for flexibility on the open time-interval $(0,1)$. The parameters of the interpolation are trained together with the those of the flow (and those of $C_t$) with the objective of minimizing the continuity loss. Fig. 8 shows that this flexibility allows a flow trained with the continuity loss to capture both modes of the distribution (18).

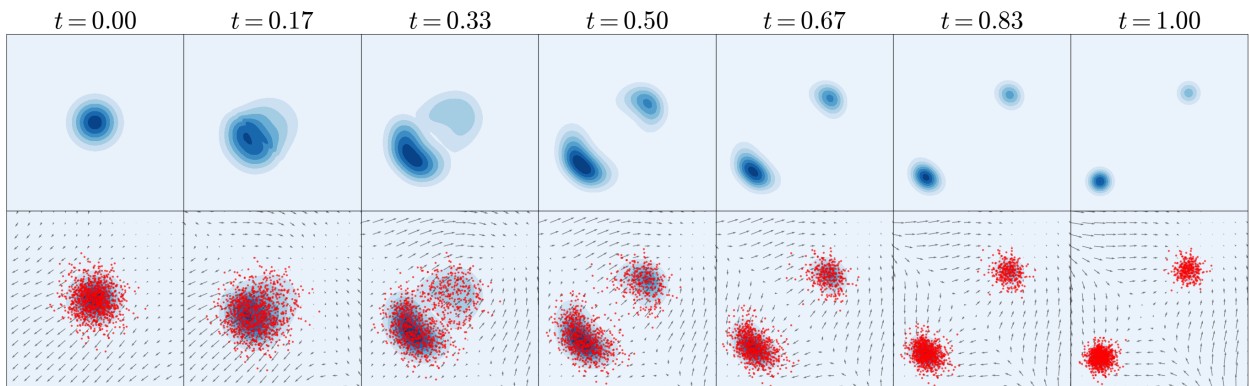

Figure 8: The same continuous normalizing flow as in Fig. 4 trained with the continuity loss using the trainable interpolation. The blue blobs denote the target density (18), a mixture of 2 Gaussians. The top row shows how the target density evolves under the learned interpolation, while the bottom row shows how the samples from $q_\theta$ evolve along $V_t$ as $t$ is varied.

## 4 Experiments

To compare the continuity loss to the reverse KL-divergence we train the same normalizing flow architecture by minimizing the 1) reverse KL objective and 2) the continuity loss. We run experiments on Gaussian mixtures and on the Boltzmann distribution of a quantum mechanical particle in a double-well potential.

**Performance metrics**

To quantify the results of the experiments, for each model we report a subset of the following metrics. For all runs we report the reverse KL-divergence (minus $\log Z$),

$$\mathbb{E}_{x \sim q_\theta}(\log q_\theta(x) - \log p(x)) \tag{21}$$

and the effective sample size,

$$ESS_r = \frac{\left(\frac{1}{N}\sum_i p(x_i)/q_\theta(x_i)\right)^2}{\frac{1}{N}\sum_i (p(x_i)/q_\theta(x_i))^2}, \qquad x_i \sim q_\theta, \tag{22}$$

where $N$ is the number of samples. These metrics capture how good a fit $q_\theta$ is for $p$, but only in those regions where samples are available. They are therefore insensitive to mode collapse. To compensate for this, we compute the Hausdorff distance between the means of the modes $M = \{m_1, ..., m_k\}$ and a batch of samples $X = \{x_1, ..., x_N\}$ from the model,

$$H(M, X) = \max_{m \in M} \min_{x \in X} \sqrt{||m - x||^2}. \tag{23}$$

In the case of Gaussian mixtures, the means $M$ are the means of mixture components whereas in the case of the quantum mechanical particle, the means are $(\phi_1, \phi_2, ..., \phi_N) = (a, a, ..., a)$ and $(\phi_1, \phi_2, ..., \phi_N) = (b, b, ..., b)$ where $a$ and $b$ are the two local minima of $V(\phi)$ (see §4.2 for details). The Hausdorff distance is a good metric for measuring mode coverage but is insensitive to the shape of the distributions. Finally, the forward KL-divergence (plus $\log Z$),

$$\mathbb{E}_{x \sim p}(\log p(x) - \log q_\theta(x)) \tag{24}$$

and effective sample size,

$$ESS_f = \frac{\left(\frac{1}{N} \sum_i q_\theta(x_i)/p(x_i)\right)^2}{\frac{1}{N} \sum_i (q_\theta(x_i)/p^\theta(x_i))^2}, \qquad x_i \sim p. \tag{25}$$

provide the most accurate (both mode coverage and shape matching) description of the goodness of the fit. As they require samples from $p$, we only report them for the experiments on Gaussian mixtures.

## 4.1 Gaussian mixtures

In this section we consider two targets, those given by (2) and (18). The metrics are reported in Table 1 and correspond well to what we observe in Figures 2, 3, 4 and 8.

Table 1: Results of training the same flow architecture with different objectives on the targets (2) and (18). Note that the target densities are normalized, i.e. $\log Z = 0$. Mean and standard deviation values over 5 seeds are reported.

| | Energy function given by log(Eq.2) | | | | |
|---|---|---|---|---|---|
| | $H(M, X) \downarrow$ | Rev. KL $\downarrow$ | Rev. ESS $\uparrow$ | Fw.KL $\downarrow$ | Fw. ESS $\uparrow$ |
| $KL(q_\theta, p)$ | $19.26 \pm .333$ | $1.387 \pm .001$ | $0.999 \pm .000$ | $110.6 \pm 97.6$ | $0.254 \pm .006$ |
| Cont. Loss with Linear Int. | $0.060 \pm .023$ | $0.003 \pm .002$ | $0.998 \pm .001$ | $0.001 \pm .001$ | $0.998 \pm .002$ |
| Cont. Loss with Trainable Int. | $0.053 \pm .010$ | $0.000 \pm .001$ | $1.000 \pm .000$ | $0.000 \pm .000$ | $0.999 \pm .000$ |
| | Energy function given by log(Eq.18) | | | | |
| | $H(M, X) \downarrow$ | Rev. KL $\downarrow$ | Rev. ESS $\uparrow$ | Fw.KL $\downarrow$ | Fw. ESS $\uparrow$ |
| $KL(q_\theta, p)$ | $13.25 \pm .363$ | $1.099 \pm .000$ | $1.000 \pm .000$ | $74.38 \pm 30.3$ | $0.332 \pm .007$ |
| Cont. Loss with Linear Int. | $2.329 \pm .124$ | $1.081 \pm .004$ | $0.928 \pm .080$ | $37.68 \pm 6.99$ | $0.336 \pm .008$ |
| Cont. Loss with Trainable Int. | $0.045 \pm .022$ | $0.001 \pm .001$ | $0.999 \pm .000$ | $0.000 \pm .000$ | $1.000 \pm .000$ |

## 4.2 Quantum mechanical particle in a double-well potential

In this section we consider the trajectory of a quantum mechanical particle moving in a one-dimensional double-well potential $V$ between $t = 0$ and $t = T$. We closely follow the experimental setup of Vaitl et al. (2022). The action associated to a continous trajectory $\phi(t) \in \mathbb{R}^n, 0 \le t \le T$ reads

$$S[\phi(t)] = \int_0^T \frac{m}{2} \left(\partial_t \phi\right)^2 + V(\phi) dt, \qquad V(\phi) = -\frac{m}{2}\phi^2 + \frac{\lambda}{4}\phi^4, \tag{26}$$

where $\lambda$ are numerical parameters. After distretizing time, the discretised action of a trajectory $(\phi_1, ..., \phi_N)$ is

$$S(\phi_1, ..., \phi_N) = \sum_{i=1}^N \left(\frac{m}{2}\left(\frac{\phi_i - \phi_{i+1}}{\Delta T}\right)^2 + V(\phi_i)\right)\Delta T, \qquad V(\phi) = -\frac{m}{2}\phi^2 + \frac{\lambda}{4}\phi^4, \tag{27}$$

where $\Delta T = T/N$, $m$ and the subscript $i + 1$ is to be understood modulo $N$. We replicate the choices of Vaitl et al. (2022) for $T = 4, \lambda = 1, N \in \{4, 8, 16, 32, 64\}$ and we use mass values $m \in \{1.50, 3.00, 4.50, 6.00\}$. The goal is then, as before, to sample trajectories $(\phi_1, ..., \phi_N)$ from the Boltzmann density

$$p(\phi_1, ..., \phi_N) = \frac{1}{Z}e^{-S(\phi_1, ..., \phi_N)}, \qquad Z = \int_{\mathbb{R}^N} e^{-S(\phi_1, ..., \phi_N)} d^N x \tag{28}$$

**Heuristics.** For larger values of $m$, the energy barrier between the wells of $V$ gets greater, resulting in a bimodal, one-dimensional Boltzmann density $e^{-V(\phi)}$. (Fig. 9). Intuitively, the second summand in (27) encourages the particle to follow the unnormalized density $e^{-V(\phi)}$ at every time step, while the first one penalizes if the values of $\phi$ at consecutive time steps differ too much (i.e. belong to different modes of $e^{-V(\phi)}$). With the above interpretation we can argue that $\phi_i$ and $\phi_{i+1}$ are likely to be close, and every $\phi_i$ is likely to belong to one of the modes of $e^{-V(\phi_i)}$. Then the density $e^{-S(\phi_1,...,\phi_N)}$ has two modes, centered at $(a, a, ..., a)$ and $(b, b, ..., b)$, where $a$ and $b$ are the local minima of $V$.

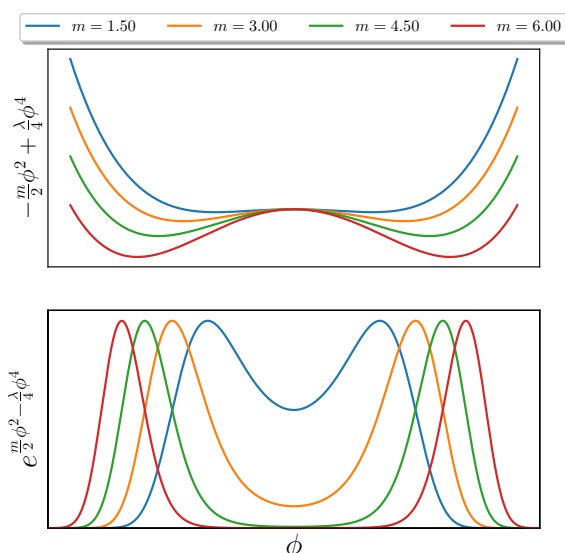

Figure 9: Dependence of $V(\phi)$ and of the Boltzmann density $e^{-V(\phi)}$ on $m$.

**Sensitivity to the mass parameter.** Now we fix $N = 16$ and vary the mass of particle $m \in \{1.50, 3.00, 4.50, 6.00\}$. We compare the continuity loss with the trainable interpolation to the reverse KL objective. The quantitative results are summarized in Table 2. In Figure 10 we compare $e^{-V(\phi)}$ to the histogram of flattened samples from the trained models. This makes sense since the action encourages the particle to follow the one-dimensional Boltzmann density of the potential $V(\phi)$ at every time step. Note that these two densities are not supposed to perfectly match, and Fig. 10 can only be used to detect mode-collapse of the flow.

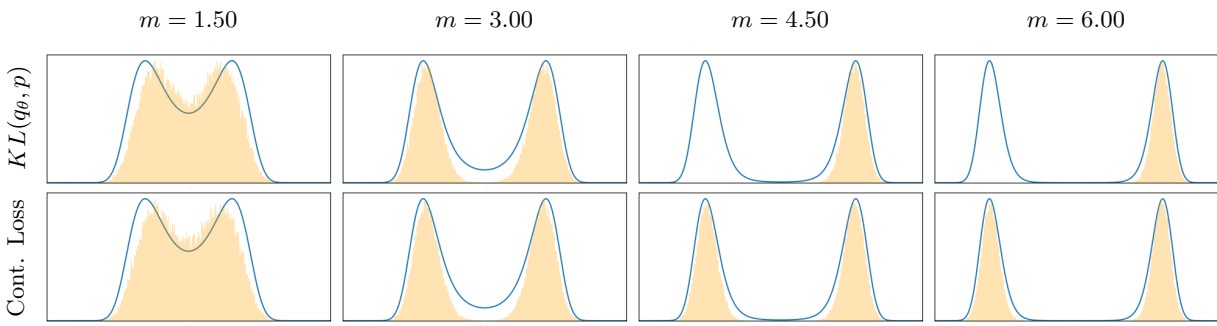

Figure 10: Mode collapse of the reverse KL-divergence for higher values of $m$. The unnormalized density $p(\phi) \propto e^{-V(\phi)}$ (blue curve), compared to flattened samples $\phi_i$ (orange histogram).

Table 2: Results of training the same flow architecture with different objectives on the energy function (27). Note that a decrease of $\log 2 \approx 0.7$ in reverse KL-divergence corresponds to covering twice as much probability mass. Mean and standard deviation values over 3 seeds are reported.

| | $m = 1.50$ | | | $m = 3.00$ | | |
|---|---|---|---|---|---|---|
| | $H(M, X) \downarrow$ | Rev. KL $\downarrow$ | Rev. ESS $\uparrow$ | $H(M, X) \downarrow$ | Rev. KL $\downarrow$ | Rev. ESS $\uparrow$ |
| $KL(q_\theta, p)$ | $0.717 \pm .06$ | $-1.052 \pm .00$ | $0.951 \pm .00$ | $0.541 \pm .00$ | $-1.832 \pm .01$ | $0.801 \pm .11$ |
| Continuity Loss | $0.678 \pm .02$ | $-1.054 \pm .00$ | $0.991 \pm .00$ | $0.486 \pm .04$ | $-1.835 \pm .00$ | $0.979 \pm .00$ |

| | $m = 4.50$ | | | $m = 6.00$ | | |
|---|---|---|---|---|---|---|
| | $H(M, X) \downarrow$ | Rev. KL $\downarrow$ | Rev. ESS $\uparrow$ | $H(M, X) \downarrow$ | Rev. KL $\downarrow$ | Rev. ESS $\uparrow$ |
| $KL(q_\theta, p)$ | $13.61 \pm .49$ | $-9.058 \pm .00$ | $0.883 \pm .05$ | $16.67 \pm .00$ | $-22.49 \pm .01$ | $0.935 \pm .01$ |
| Continuity Loss | $\mathbf{0.335} \pm .00$ | $\mathbf{-9.765} \pm .01$ | $0.966 \pm .01$ | $\mathbf{0.300} \pm .00$ | $\mathbf{-23.19} \pm .01$ | $0.963 \pm .01$ |

**Sensitivity to the dimensionality and computational speedup.** In this section we fix a relatively small mass value, $m = 1.50$, resulting in a unimodal Boltzmann density. We then train models at different time resolutions, $N \in \{4, 8, 16, 32, 64\}$. For each $N$ we trained two models, one with the reverse KL and one with the continuity loss. Figure 11 shows the evolution of the (reverse) effective sample size and the continuity loss during the training process. Since the continuity loss is a pointwise objective, contrary to the reverse KL, it does not require backpropagation along the trajectories of the flow. On the other hand, the "baseline" reverse KL objective only requires a parametrization of $V_t$ and the computation of $\nabla \cdot V_t$. In addition to this, the continuity loss also requires to parametrizations of $C_t$ and $f_t$ and the computation of $\partial_t f, \nabla f$ and $C_t$. Overall, we observed a speedup when switching to the continuity loss both in terms of number of optimization steps per unit time and also in terms of convergence speed (Figure 11).

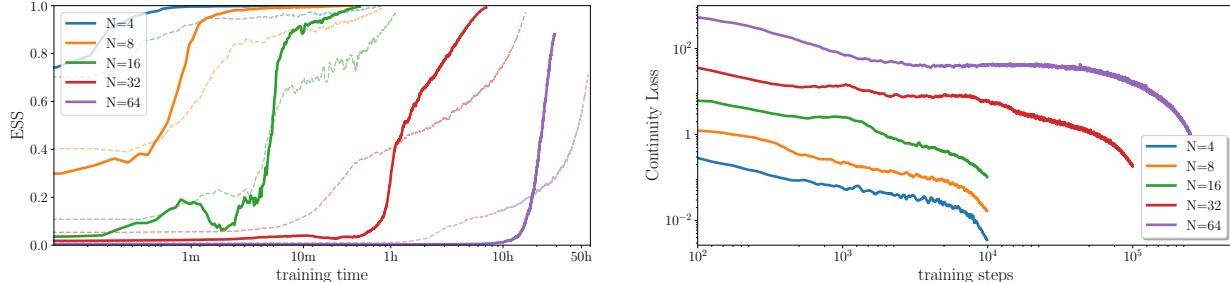

Figure 11: Evolution of the (reverse) effective sample size (left) and of the continuity loss (right) during training for different dimensionality of the problem. The solid lines correspond to the models trained by optimizing the continuity loss, while the dashed lines denote the models trained by optimizing the reverse KL divergence.

## Implementation and training details

**Architectures.** The parametrization of $V_t$ and $f_t$ are given by a weighted average of 4 MLPs. The weighting is done by evenly spaced RBF time-kernels, one for each model. In the case of the Gaussian targets the MLPs has two hidden layers with 64 neurons per layer, in the case of the quantum particle the MLPs has 3 hidden layers with 128 neurons per layer. Between the linear layers we use swish nonlinearities. The parametrization of $C_t$ consists of a single MLP with the same hyperparameters as the mixture components of the parametrizations of $V_t$ and $f_t$. Importantly, our architecture is completely oblivious to the $\mathbb{Z}_2 \ltimes C_n$-symmetry of the lattice of quantum mechanical particle and to the $\mathbb{Z}_2$-symmetry of the double-well potential. We rely on automatic differentiation to compute all derivatives. We leave the exploitation of symmetries and the use of architectures with analytic expressions for the divergence of $V_t$ (Köhler et al., 2020; Gerdes et al., 2022), as well as for $\nabla f_t$ and $\partial_t f_t$, for future work.

**Optimization.** We train with a batch size of 256 using the Adam optimizer (Kingma & Ba, 2017) and evaluate on batches of size 4096. The trajectories of the flow are computed by a 4th-order Runge-Kutta solver with 50 integration steps. The $N = 64$ and $N = 32$ runs in §4.2 are trained for $2.5 \times 10^5$ and $10^5$ iterations, respectively. All other models are trained for $10^4$ iterations. The initial learning rate of $3 \times 10^{-3}$ is annealed to 0 following a cosine schedule.

**Base Density.** We use a standard Gaussian base for the target (2), a centered Gaussian with standard deviation 2 for the target (18) and the generalized Gaussian proportional to $e^{-x^4}$ in the experiments of §4.2.

**On the choice of the function $L$.** The definition of the the continuity loss (16) involves a somewhat arbitrary choice of the function $L$. We compared the functions $\{\mathcal{E} \mapsto \mathcal{E}, \mathcal{E} \mapsto \mathcal{E}^2, \mathcal{E} \mapsto \mathcal{E} + \mathcal{E}^2, \mathcal{E} \mapsto \log(1 + \mathcal{E})\}$ and ended up using $\mathcal{E} + \mathcal{E}^2$ in all of our experiments, as it empirically outperformed the other choices. The intuition is that the $L^2$ norm provides good gradients when the error is large, while the $L^1$ norm provides

good gradients when the error is small and therefore their sum is expected to be superior to both of them. Our experiments support this intuition.

## 5 Summary and Closing remarks

We introduced an alternative training objective of continuous normalizing flows that uses an interpolation of energy functions. We've demonstrated empirically that the proposed objective is less prone to mode collapse than the KL-divergence when the target density has multiple modes and is computationally more efficient.

**Two reasons for mode collapse.** There can be, at least, two different reasons for mode collapse when training with the reverse KL-divergence. First, if one of the modes is so far away from the base distribution that it never gets visited by the flow. Second, if a mode is visited during training by trajectories of the flow, but the flow still ignores it and fits the remaining modes of the density. It is important to distinguish these two scenarios as our proposed technique can help with the second kind, but not the first one. This is also the reason why the choice of the base density is important. When the base has a higher variance, a larger fraction of the space gets visited.

**Reframing the method as a PINN.** Our work naturally fits into the framework of Physics-informed neural networks (Raissi et al., 2019). What this work calls the pointwise continuity error, would be called the residual to the continuity equation in the PINN literature. The core idea is essentially the same: the optmization of a neural network to satisfy a PDE.

## 6 Acknowledgement

The authors acknowledge support from the Swiss National Science Foundation under grant number CR-SII5_193716 - "Robust Deep Density Models for High-Energy Particle Physics and Solar Flare Analysis (RODEM)". We also thank Samuel Klein, Jonas Köhler and Eloi Alonso for discussions, Saviz Mowlavi for mentioning the connection to PINNs, and Ricky T. Q. Chen for pointing out that (12) is just the continuity equation.

We would also like to express our appreciation of the Python ecosystem (Van Rossum & Drake Jr, 1995) and its libraries that made this work possible. In particular, the experimental part of the paper relies heavily on JAX (Bradbury et al., 2018), haiku (Hennigan et al., 2020), optax (Babuschkin et al., 2020), numpy (Harris et al., 2020), hydra (Yadan, 2019), matplotlib (Hunter, 2007), jupyter (Pérez & Granger, 2007) and weights&biases (Biewald, 2020).

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
