# OpenReview forum: "Learning Interpolations between Boltzmann Densities"
_TMLR — Accepted by TMLR_

### Review · Reviewer_t7Z6 · 2023-02-13

**Summary Of Contributions:**

The authors introduced a new method of approximating a normalizing flow based on interpolating the energy functional. With experiments the authors showed that the method avoids collapsing on a single mode.

**Audience:**

Yes

**Claims And Evidence:**

Yes

**Requested Changes:**

To be written later based on the answers I receive.

### Update: Feb 23

I would like the authors to address the my comments in the weaknesses section by providing
1. A comparison with alternative methods, and a discussion on how to judge between the different methods.
2. Some additional experiments and discussion on how the performance or implementation difficulty changes depending on the parameters of the given distribution.

**Strengths And Weaknesses:**

Since I'm not an expert on the subject of normalizing flows, I would like to use this opportunity to ask the authors several possibly basic questions, before providing a review judging the strengths and weaknesses. I will return and modify this section as well as the request changes section based on the answers I receive.

The most important questions I have are about the training procedure. Firstly, can the authors provide a more clear description of the model that is being trained all in one place? I have a lot of confusion over this at the moment, and in particular, I would like address several of my detailed questions below:
1. Is the restriction to training only interpolated models the only mechanism ensuring the flow reaches the right target?
2. As a result, is the training loss essentially just ensuring the interpolation still forms a flow, where $V_t^\theta, C_t^\theta, f_t^\theta$ are compatible? In particular, isn't $C_t$ determined by $f_t$ already, since it's arising from the normalizing constant?
3. How is $q_\theta$ parameterized by $V_t^\theta$, and what is this object?
4. What is $S$ in the definition of $w(z)$?
5. I'm not an expert on the subject, but is the target $f_1$ always known for problems of this type?

### Update: Feb 23

I want to thank the authors for providing quick answers to my questions. I believe at this point I have a better understanding of the paper, and I will provide a review in terms of the strengths and weaknesses. However, I will still refrain from providing a confident judgement of the paper, as I am not an expert on the subject of normalizing flows.

Strengths
1. The authors provided a new method for sampling using a flow based on technique, that is trained to ensure the flow structure is maintained.
2. The approach avoids the explicit calculation of normalizing constants along the trajectory.
3. Experiments show good behaviours of the algorithm.

Weaknesses
1. Perhaps it's just due to my lack of knowledge of this field, but I don't know how to compare the method the authors introduces to other comparable approaches.
2. It's unclear how the parameters affect the computational difficulty of the method, e.g. the dimension of the problem, the distance between the different modes.

---

> ### Author Response · Authors · 2023-02-20
> **Response to  Reviewer t7Z6**
>
> Thank you for your questions!
>
> 1) We are aware of two other approaches (Vaitl et al. (2022) and Midgley et al. (2022)) tackling the same problem.
>
> 2) Exactly, the training is essentially just ensuring that $V_t^\theta,C_t^\theta,f_t^\theta$ are compatible. Indeed, $C_t$ is determined by $f_t$ already in theory, but it's intractable to compute in practice. To this end, we approximate it with a neural network.
>
> 3) $q_\theta$ is a probability density function induced by the continous normalizing flow. In abstract terms this means that $q_\theta$ is the density obtained by a change of variables $q_\theta(x) = p_0(\Phi^{-1} x) |\det \Phi^{-1}(x)|$, where $p_0$ is the base density and $\Phi$ denotes the diffeomorphism of following the integral curves of $V_t$ between $0$ and $1$. In practice this means that we can sample from $q_\theta$ by first sampling from the base $p_0$ and making the samples follow the integral curves of $V_t$ between $0$ and $1$.
>
> 4) This was a typo, $S$ is now replaced with $f_1$ in the definition of $w(z)$ as well as in Eq. 5. Thank you for spotting it!
>
> 5) Exactly, the target energy function $f_1$ is known.

---

> > ### Comment · Reviewer_t7Z6 · 2023-02-21
> > **Brief Follow-Up**
> >
> > Thank you for the quick answers, these are very helpful. Just some quick follow-up questions before I start writing a review.
> >
> > Firstly, I want to get a deeper interpretation of the loss function $\mathcal{L}(\theta)$ in equation 9. Here are some specific questions:
> > 1. What is the base distribution $\mathcal{B}$?
> > 2. Can you provide more intuitions on why this weighting by $w(z)$ should help?
> >
> > Secondly, in your motivation, you described the mode-seeking nature of KL-divergence. However, if you are strictly limiting your method to interpolation with the knowledge of the target density, don't you automatically rule out the possibility of seeking modes? This might be yet another dumb question given I'm not an expert on the subject, but is this really a problem if you already know the target?

---

> > > ### Author Response · Authors · 2023-02-23
> > > **Brief Follow-Up**
> > >
> > > Thank you for the questions!
> > >
> > > 1. The base density is what we referred to as $p_0$ in the previous response, i.e. a density that we can sample from and evaluate the likelihood of. The task of the flow is then to find an invertible map between this base and the target/data distribution. The base is usually a Gaussian, but in some of our experiments we found that we had better results with generalized Gaussians.
> > >
> > > 2. The weights $w(z)$ are essentially importance weights, i.e.they reweight the trajectories from the likelihood of its endpoint under $q_\theta$ to the its (unnormalized) likelihood under $e^{-f}$.
> > >
> > > When we say that we know the target, we mean that we can evaluate its unnormalized density $e^{-f}$ anywhere, but we have no practical way of sampling from it. The task is then to leverage the former to achieve the latter.

---

> > > > ### Author Response · Authors · 2023-03-15
> > > > **Response to the Feb 23 update of Reviewer t7Z6**
> > > >
> > > > Thank you for your review and for going through the paper even though you’re not an expert in the field!
> > > >
> > > > >Some additional experiments and discussion on how the performance or implementation difficulty changes depending on the parameters of the given distribution.
> > > >
> > > > We added some experiments to understand the sensitivity of the method to the dimensionality of the problem.
> > > >
> > > > >A comparison with alternative methods, and a discussion on how to judge between the different methods.
> > > >
> > > > We implemented one of the alternative methods and started experiments on the target densities we used in our paper. The results on the two-dimensional Gaussians are now in the updated version of the paper. The larger-scale experiments are currently running, we will add the results once they are done. With this being said, we would also like to emphasize that the methods trying to replace the reverse KL divergence are all relatively new (the ones we are aware of were published at ICML 2022 and ICLR 2023). It is not yet clear what the advantages and disadvantages of either of these methods are compared to the others, and an experimental comparison would likely be more meaningful after all methods have been better understood and tweaked to their best forms.
> > > >
> > > > ### Update Mar 20
> > > >
> > > > 1) All the experimental results are now included in the paper.
> > > >
> > > > 2) Regarding our earlier exchange about the importance weights $w(z)$. Due to a broadcasting error, all the trajectories were weighted with 1 in our implementation, i.e. they were not weighted relative to each other. The updated paper presents the simplified version of the training objective not containing $w(z)$.

---

### Review · Reviewer_vNnM · 2023-02-23

**Summary Of Contributions:**

The paper addresses sampling from an unnormalised model by fitting a continuous-time projection/action that pushes sample from an initial ( generalised Gaussian) distribution to the target energy function. The main contribution is the particular interpolation scheme, which is a combination of the usual linear interpolation between simple and target energies, and a novel a transient additive component that helps transporting the samples across different time steps.

I apologise that my review is brief as I have very limited time, but I have understood the paper and please bear with me if anything is unclear.

**Audience:**

Yes

**Broader Impact Concerns:**

Not relevant

**Claims And Evidence:**

No

**Requested Changes:**

## Critical:
* Address my first weakpoint and answer the question: why not using diffusion?
* Above eqn 8, the functions in footnote are not monotonically increasing.
* Page 6, first para.: The issue isn't purely optimizability, Langevin dynamics would also work, and initialisation also matters.
* Page 6, I don't see the purpose of the sanity check section, apart from saying: we don't want to evaluate gradients so don't use diffusion. Please clarifty the point here.
* Page 6 eqn 12. The symbols St and Ct are overloaded, please change them.
* Experiments should include some score-based methods (e.g. SVGD by Liu & Wang 2016), unless the example target density is non-differentiable, a case the authors seem to focus on, even if the current approach underperforms than this method.

## Good to have
1. Performance metrics should include maximum mean discrepancy (Gretton et al, 2012)
2. Section 4.2 needs to be made friendly to more general machine learning audience, as well as an earlier mention of phi4 theory. If (19) isn't used later for experiments, don't write it down, just present (21). The intermiate discussions are very confusing for general ML readers.

**Strengths And Weaknesses:**

The paper gives a very clear motivation and introduction of fitting a normalise flow model to a unnormalised density for the purpose of sampling. The examples are simple to understand, but perhaps too toy-ish for machine learning community. The main contribution is sound and I think it's simple but effective.

The weakensses are mainly the demonstrated significance of the idea, and some clarify.
* One assumption in this work is that the gradient of the energy is unavailable, but no examples have shown this application. Does such scenario arise in physics? If so, show one, otherwise just use diffusion.
* The main application is the phi-4 theory, which is not presented very clearly.
* Relation to existing literature: some issues discussed in this work are related to energy-based or score-based methods in machine learning, and I encourage the authors to at least acknowledge the literature in writing (e.g. Song and Ermon 2019, Wenliang & Kanagawa 2021).
* There are some design choices regarding the loss function, and there is no experiments showing the effect of this choice.
* It would be great if one can demonstrate an example on moderate dimensional densities.

---

> ### Author Response · Authors · 2023-03-15
> **Response to Reviewer vNnM**
>
> Thank you for taking the effort to review despite the limited amount of time you have!
>
> >One assumption in this work is that the gradient of the energy is unavailable, but no examples have shown this application. Does such scenario arise in physics? If so, show one, otherwise just use diffusion.
>
> We actually assume that the gradient of the energy is available. In fact we work with a smooth family of energy functions that we compute the derivatives of (both in time and space) every training step (Eq. 8).
> Regarding diffusion, if  you mean to use denoising diffusion models, we cannot, as they rely on having samples from the target density. If you mean to use the interpolation along a predefined diffusion path, we avoided this because this diffusion path is expensive to compute for general densities.
>
> >The main application is the phi-4 theory, which is not presented very clearly.
>
> This section is now rewritten, we believe that the presentation is now cleaner.
>
> >Relation to existing literature: some issues discussed in this work are related to energy-based or score-based methods in machine learning, and I encourage the authors to at least acknowledge the literature in writing (e.g. Song and Ermon 2019, Wenliang & Kanagawa 2021).
>
> We agree, the ideas of this work were motivated by denosing diffusion and score-based models, which is now properly acknowledged in the updated version of the paper.
>
> >Above eqn 8, the functions in footnote are not monotonically increasing.
>
> We updated the paper to fix this.
>
> >Page 6, first para.: The issue isn't purely optimizability, Langevin dynamics would also work, and initialisation also matters.
>
> We changed the phrasing.
>
> >Page 6 eqn 12. The symbols St and Ct are overloaded, please change them.
>
> Good catch, we updated the paper to fix this.
>
> >Experiments should include some score-based methods (e.g. SVGD by Liu & Wang 2016), unless the example target density is non-differentiable, a case the authors seem to focus on, even if the current approach underperforms than this method.
>
> The issue with SVGD is that it provides samples that in the limit of infinite iterations  follow the target distribution, but it does not provide a density function to evaluate. Having access to the density is critical to correct  the learnt model, and use it as a proposal distribution in importance sampling or in MCMC schemes. This was not clear from the original version of the paper, we updated the introduction with this explanation.

---

### Review · Reviewer_Hzj2 · 2023-03-08

**Summary Of Contributions:**

This paper proposes a method to sample from the target distribution proportional to $\exp(-f_1)$, where $f_1$ is a real valued function defined over a vector space.

The method relies on first sampling from an easy distribution $\exp(-f_0)$ (e.g., a Gaussian) and then transporting the samples to the target.
To transport the samples to the target, the authors propose to consider a path $f_t, t \in [0,1]$ from $f_0$ to $f_t$ and the associated distribution $p_t \propto \exp(-f_t)$. Since $p_t$ joins the easy distribution $\exp(-f_0)$ to the target $\exp(-f_1)$, the method relies on learning a velocity field $v_t$ of $p_t$, then the samples can be transported by solving the differential equation $x'(t)= v_t(x(t))$.

The main contribution of the paper is experimental: they show that the proposed method addresses the mode collapse behavior of concurrent methods based on optimizing the reverse KL. The proposed method is therefore more suitable when $f_1$ is not convex.



**Audience:**

Yes

**Claims And Evidence:**

Yes

**Requested Changes:**

- Address the weaknesses mentioned above. These are mainly clarifications.

- Define velocity field formally. I also wanted to establish Eq (4) from the definition of velocity field but I could not. Could you explain why Eq 4 holds?

- Before the summary of contributions, explain what the advantages of the method are.

- Rewrite the abstract.

- Explain why an approximation is needed in Eq (21).

Some questions:

- When I saw Eq (7), I wanted to take the gradient to get rid of $C_t$. This new equation could be taken as the PDE. Any thoughts?

- Instead of learning $f_t$, could we learn $\nabla f_t$? This vector is known as the score and there are efficient methods to learn the score that were successfully used in the field of diffusion models.




**Strengths And Weaknesses:**

Strengths:

- The paper is well written and pleasant to read, I liked to see controlled experiments to strengthen the story before section 4.

- My personal opinion is that the overall approach makes sense theoretically.

- There are real improvements on the practical side.


Weaknesses:

- There is no theoretical justification/ proofs on why the proposed method should work. Maybe the authors could consider a toy model for the parametrization and explain the behavior of the algo in a simple case? Or, maybe write a paragraph to explain why intuitively the proposed method should not be mode seeking (for example an intuitive explanation of what we observe in Eq 3)?

- There is no discussion on how the ODE is discretized once the velocity field is learned.

- The path joining $p_0$ to $p_1$ seems arbitrary, isn't there a more canonical one? Even the path that is learned (Eq 13) does not seem canonical to me.

- The abstract is not well written. Too many uses of the word "this", it is not always clear what it refers to. Besides, "minimize the amount ..." is not clear at this stage of the reading.

- Related to the above, in the deformation loss, the choice of L and the weights is not discussed.

---

> ### Author Response · Authors · 2023-03-15
> **Response to Reviewer Hzj2**
>
>
> Thank you for your review and for raising some interesting ideas!
>
> >There is no theoretical justification/ proofs on why the proposed method should work. Maybe the authors could consider a toy model for the parametrization and explain the behavior of the algo in a simple case? Or, maybe write a paragraph to explain why intuitively the proposed method should not be mode seeking (for example an intuitive explanation of what we observe in Eq 3)
>
> One intuitive explanation for the observed behavior of the algorithm is the “continuous reward signal” provided by the continuous family of targets as opposed to the “once per trajectory reward” provided by the optimization of the reverse KL. We describe this in the beginning of section 3. We also updated Figures 5 and 6 to show the norm of the transport field which contributes to the understanding of the issue with the linear interpolation.
>
> >There is no discussion on how the ODE is discretized once the velocity field is learned.
>
> We are not sure what you mean by this. Could you say a bit more?
>
> >The path joining $p_0$ to $p_1$ seems arbitrary, isn't there a more canonical one? Even the path that is learned (Eq 13) does not seem canonical to me.
>
> We agree that these choices are somewhat arbitrary. There are alternative choices one could take, none of which seems canonical. Our choice was a simple one that served the purpose of demonstrating the feasibility of the method. Most likely, if we want to push the technique to its boundaries, the choice of the interpolation path requires more care.
>
> >Related to the above, in the deformation loss, the choice of L and the weights is not discussed.
>
> This is now included in the updated version of the paper.
>
> >Define velocity field formally. I also wanted to establish Eq (4) from the definition of velocity field but I could not. Could you explain why Eq 4 holds?
>
> I assume that by velocity field  you mean the vector field $V_t$. We follow both for $V_t$ and Eq (4) the same notions as in the Neural ODE literature (1806.07366, Eq. 8).
>
> >The abstract is not well written. Too many uses of the word "this", it is not always clear what it refers to. Besides, "minimize the amount ..." is not clear at this stage of the reading.
>
> We have adjusted the abstract taking your comments into account.
>
> >Explain why an approximation is needed in Eq (21).
>
> We simplified section 4.2, this approximation is not part of the paper anymore.
>
> >When I saw Eq (7), I wanted to take the gradient to get rid of $C_t$. This new equation could be taken as the PDE. Any thoughts?
>
> This is something we considered as it would indeed mean getting rid of $C_t$, but taking the gradient of Eq (7) at every training step seems to be computationally more expensive than having an additional model for a $[0,1] \rightarrow \mathbb{R}$ function.
>
> >Instead of learning $f_t$, could we learn $\nabla f_t$? This vector is known as the score and there are efficient methods to learn the score that were successfully used in the field of diffusion models.
>
> This is an interesting idea! This would create a more direct connection to score-based models. A difficulty would be that given a parametrization of $\nabla f$, $\partial_t f$ is not as straightforward to obtain. Thank you for this point, we will need to think about it carefully.

---

> > ### Comment · Reviewer_Hzj2 · 2023-03-24
> > **Thanks for the reply**
> >
> > - After the learning phase, samples are generated by generating a simple distribution (say, a Gaussian) and then solving an ODE, right? How do you solve the ODE? With Euler method?
> >
> > Overall, I am happy with the revision. I feel that the connection with score based methods could be exploited (maybe in future works).

---

> > > ### Author Response · Authors · 2023-03-27
> > > **ODE-solver**
> > >
> > > > How do you solve the ODE? With Euler method?
> > >
> > > Thank you for the clarification! We use a 4th-order Runge-Kutta solver, which is now also included in the section discussing the implementation details.

---

### Decision · Action_Editors · 2023-05-25

**Recommendation:** Accept as is

**Comment:**

The paper proposes a new method for approximating a normalizing flow based on interpolating the energy functional. The proposed method addresses the issue of collapsing to a single mode in normalizing flow models and has been shown to perform well in experiments. The method involves fitting a continuous-time projection/action that pushes samples from an initial distribution to the target energy function using a combination of linear interpolation and a novel transient additive component. There were some minor concerns and the revision has addressed all concerns. So I recommend accept.



**Audience:**

Yes. Would be of interested to researchers working on normalizing flows.

**Claims And Evidence:**

Yes. The paper proposes a new method for approximating a normalizing flow based on interpolating the energy functional. The proposed method claims to address the issue of collapsing to a single mode in normalizing flow models, and the experiments are convincing.